# Sudden Cardiac Death Risk over Time in HCM Patients with Implantable Cardioverter-Defibrillator

**DOI:** 10.3390/jcm11061633

**Published:** 2022-03-16

**Authors:** Mariusz Klopotowski, Krzysztof Kukula, Jacek Jamiolkowski, Artur Oreziak, Maciej Dabrowski, Zbigniew Chmielak, Adam Witkowski

**Affiliations:** 1Department of Interventional Cardiology and Angiology, Cardinal Wyszynski National Institute of Cardiology, 04-628 Warsaw, Poland; krzysiokuk@yahoo.com (K.K.); mdabrowski@ikard.pl (M.D.); zchmielak@ikard.pl (Z.C.); witkowski@hbz.pl (A.W.); 2Department of Population Medicine and Civilization Disease Prevention, Medical University of Bialystok, 15-269 Bialystok, Poland; jacek909@wp.pl; 3Department of Cardiac Arrhythmias, Cardinal Wyszynski National Institute of Cardiology, 04-628 Warsaw, Poland; aoreziak@ikard.pl

**Keywords:** hypertrophic cardiomyopathy, implantable cardioverter-defibrillator, sudden cardiac death, risk

## Abstract

Background: In patients with HCM at high risk of SCD, an ICD should be considered as a standard of care. Current risk approximation algorithms recommended by ESC 2014 criteria indicate that SCD risk is not stable. The aim of the study was to investigate how the calculated SCD risk in HCM patients with an ICD implanted in the past changed over time. Methods: We analyzed 64 HCM patients with ICD for primary prevention, referred for ICD re-implantation, and 32 HCM patients referred for a first-time ICD placement during the same period. The 5-year-SCD risk was assessed for suitable patients using the recommended ESC calculator. Results: The first-time group had a higher 5-year-SCD risk than those referred for ICD re-implantation: 7.50 (IQR 5.98–10.46) vs. 4.88 (IQR 3.42–7.25), *p* < 0.05. Out of the patients with an initial calculated risk below 4%, the risk increased in 22% of cases, reaching the 4–6% range. In 78% of patients, the risk remained stable and low. In 31% of patients with an initial calculated SCD risk ≥ 6%, the risk decreased over time to below 6%, and in 14% of the cases, below 4%. Conclusions: SCD risk in HCM patients is usually stable or gets lower. Our data suggest it is important to re-evaluate the risk profile for patients with HCM when ICD re-implantation is considered.

## 1. Introduction

Patients with hypertrophic cardiomyopathy (HCM) are known to be at risk of sudden cardiac death. Estimating this risk is vital in order to treat patients appropriately, as it varies widely depending on a number of factors. While in most HCM patients the annual risk of sudden cardiac death (SCD) is below 1%, many patients are burdened with a much higher risk [1]. Hence, one of the most important clinical decisions in (HCM) patients is whether or not to offer them an implantable cardioverter-defibrillator (ICD) for primary sudden cardiac death (SCD) prevention. Recent data show that selected high-risk patients—those who have already had an ICD implanted in the past for primary prevention, have a cumulative 5-year probability of an appropriate ICD intervention of 10.5% (95% CI, 8.0–13) [2].

Recommendations for ICD implantation have evolved with the elucidation of specific clinical and echocardiographic SCD risk factors [3,4,5,6,7,8]. In spite of that, while it is indisputable that secondary SCD prevention is an established indication for ICD, primary prevention indications are still not uniform [6,8].

An ICD device requires periodic replacement due to battery depletion, failure, technical issues, or infectious complications. It creates a significant medical and economic burden. In contemporary clinical practice, the group of patients in whom an ICD needs to be replaced grows quickly over time. However, SCD risk may also change over time and some patients, upon reassessment, may no longer have indications of an ICD. In the case of infectious complications, it is advocated that indications for repeat implantation be reassessed, but there are no such recommendations in case of battery depletion [9].

The current risk approximation algorithm recommended by the ESC 2014 criteria indicates that SCD risk is not stable [6]. This calculator tool, despite its limitations, is often used to quantitatively present the 5-year SCD risk of HCM patients and was also used in this study to quantitatively express the SCD risk in our cohort.

The aim of the study was to investigate how the SCD risk in HCM patients with an ICD implanted in the past changed over time.

## 2. Materials and Methods

We compared the same cohort of HCM patients at 2 points in time: (1) historically, when they underwent ICD implantation in the past for the first time; (2) recently, when between 2014 and 2017, they developed indications for an ICD replacement due to battery depletion or device failure. A separate analysis was undertaken to compare the risk profile of this group of patients to a contemporary group with indications for a first ICD placement.

Despite its limitations and less than universal acceptance, in order to quantify a SCD risk, we used the ESC-advocated risk calculator. It is used in our center as a tool we refer to, although, of course, full individual patient risk assessment encompassing more factors than those included in the calculator is always performed [10]. However, as the score gives a numerical risk range, as with any validated score or index, it is invaluable for quantification purposes.

Patients with HCM under clinical care of the National Institute of Cardiology in Warsaw were retrospectively evaluated during their scheduled annual visit. This hospital serves as a national referral center for patients with HCM. We conducted an analysis of 91 HCM patients who underwent an ICD implantation between 2005 and 2013 for primary prevention and in whom a potential indication for device replacement arose, including battery depletion, device malfunction, and infection. Those from this group who still fulfilled the criteria of evaluation were re-evaluated, with consideration to the new guidelines. Patients were evaluated between September 2014 and April 2017.

Additionally, 36 HCM patients referred for de novo ICD placement during the same period were analyzed and served as a comparison group.

### 2.1. Risk Calculation and Comparisons

According to the SCD HCM risk calculator, the following HCM patients were excluded from the final analysis (11):History of aborted sudden cardiac death or sustained VT;With a maximum left ventricular wall thickness ≥ 35 mm;After an invasive reduction in LVOT obstruction;Known metabolic disease or syndrome.

After applying the exclusion criteria, the ESC risk calculator could be used for risk assessment in 64 patients referred for ICD re-implantation for primary prevention and 32 de novo ICD implantation group. Additionally, in the same patients, we calculated the SCD risk at the time of the first implantation. All the necessary data were available in clinical records. Next, we compared patients’ risk before the first ICD implantation and at the time of device replacement. We also compared the SCD risk to the risk of patients in the de novo ICD implantation group—those currently being referred for a first-time ICD implantation.

### 2.2. Statistics

The empiric distribution of the risk of SCD at 5 years was compared to the normal distribution using the Shapiro–Wilk test. As the data distribution significantly diverged from a normal distribution, nonparametric methods were used in data analysis. Data were presented as medians and quartiles. Risk comparison between the study cohort at 2 time points and the control cohort was done using the Mann–Whitney test. The risk difference at the 2 time points in the study cohort was assessed via the Wilcoxon matched-pairs test. The risk level was also categorized, and assessment was performed for thus created categorical variables. The distribution of risk levels between the study group at 2 time points and the control group was tested with the chi-square Pearson test and the differences in risk categories between the 2 assessments in the study group were assessed via the McNemar–Bowker test. All calculations were done using the IBM SPSS Statistics 20 software. The significance level was set to 0.05.

## 3. Results

The clinical course of HCM patients referred for ICD replacement and analysis of cases excluded from quantification using the ESC calculator.

We analyzed a group of 91 patients implanted with ICD between 2005 and 2013 for primary prevention. In 70 patients it was the first device placement, and in 21, the second or third. All these patients were referred for device replacement. In 84 patients, the indication for replacement was battery depletion, in 6 cases electrode malfunction, and in 1 case—subclavian vein thrombosis. 

The mean time from the first implantation to device replacement was 7.1 ± 2.7 years. 

Prior to the first ICD implantation for primary prevention, seven patients had undergone an alcohol septal ablation procedure and one—a surgical myectomy with mitral valve replacement. After ICD implantation, 12 other patients underwent an invasive reduction of left ventricle outflow obstruction (8 patients had surgical myectomy and 4 patients had alcohol septal ablation). Massive hypertrophy (36 mm septum thickness) was present during ICD implantation in 1 patient. They were excluded from the ESC risk score calculation and analysis.

Since the first ICD implantation, eight patients experienced an appropriate device discharge (among them one patient post myectomy with mitral valve replacement and one patient with massive hypertrophy), hence device replacement was indicated for secondary SCD prevention in those cases. They were excluded from the analysis. 

Two patients referred for device replacement refused the procedure due to very low 5-year SCD risk at their reassessment. The ICD re-implantation was scheduled for them due to battery depletion. Interestingly, their 5-year SCD risk was 2.01% and 3.26% at the moment of implantation and 1.82% and 2.63% when referred for re-implantation. Both patients had had ICDs implanted for nsVT episodes registered for 24-h Holter examinations (based on previous guidelines).

### 3.1. De Novo ICD Implantation

In the group of 36 patients with first-time implantation 3 patients had a prior alcohol septal ablation and 1 had undergone surgical myectomy, so the ESC calculator was not used in those patients.

### 3.2. ICD Appropriate Therapy

In patients from the re-implantation cohort who had had an ICD implanted for primary SCD prevention, an appropriate therapy occurred in eight patients, and in a further four after device replacement.

In the de novo ICD implantation group, appropriate therapy occurred in two cases.

Overall, there were 14 appropriate ICD discharges observed in patients referred for ICD implantation for primary prevention. Their risk factors and risk calculation are presented in Table 1.

The 5-year SCD risk in those patients was generally high. In all of them, the 5-year SCD risk was higher than 6%.

### 3.3. An Average SCD Risk Comparison between ICD Re-Implantation Group and for De Novo Implantation Group

After exclusion of patients with an invasive reduction of left ventricle outflow obstruction, massive hypertrophy, and those who have had ICD appropriate therapy after a primary implantation, 64 patients (mean age 43 ± 14 years, 58% male) in the re-implantation and 32 (47 ± 14 years, 59% male) in the de novo implantation group were analyzed. The calculated 5-year SCD risk is presented in Figure 1.

Interestingly, a SCD risk for HCM patients after an average 7 year post initial implantation remains medium-high and similar: 5.16 (3.72–7.44) vs. 4.88 (3.42–7.25), respectively (*p* = 0.129) (Figure 1).

Patients in whom an ICD was implanted for the first time after new ESC guidelines have been published—the de novo group—had a higher 5-year calculated SCD risk than those referred for ICD re-implantation: 7.50 (5.98–10.46) vs. 4.88 (3.42–7.25) (*p* < 0.05).

### 3.4. Analysis of SCD Risk According to the Number of Patients in a Specific Risk Category

In the next step, we analyzed the re-implantation and de novo ICD group according to the number of patients in a specific risk category. Patients were divided into the following 5-year SCD risk groups: <4% (low), 4.00–5.99% (medium) and ≥6% (high).

At the time of first implantation, 28.1% (*n* = 18) of patients had a SCD low risk, 26.6% (*n* = 17) had a medium risk and 45.3% (*n* = 29) had a high risk. At the time of re-implantation, 32.8% (*n* = 21) of patients had a SCD risk of <4%, 26.6% (*n* = 17) had a medium risk and 40.6% (*n* = 26) had a high risk. 

The majority of patients remained in the same risk category (78% in low, 47% in medium, and 69% in high) over time.

Interestingly, 12 patients decreased their risk category (3 from medium to the low category, 5 from high to medium, 4 from high to low). Moreso, 10 patients increased their risk category over time (4 from the low to medium category and 6 from medium to high). 

No patient in the de novo cohort had a calculated SCD risk below 4%, and in 8 patients, the risk was in the range of 4–5.99%, and in 24 ≥ 6%.

## 4. Discussion

Our study showed that according to the ESC 2014 criteria, at the time for re-implantation, more than half of the patients had a low or intermediate 5-year SCD risk and would have only weak indications for an ICD.

Another fact is that the 5-year SCD risk in patients with HCM currently referred for ICD implantation for primary prevention as per the ESC 2014 criteria is higher for the majority of patients.

Appropriate ICD shocks in the analyzed group of patients occurred exclusively in the highest risk patients. This would mean that low-risk patients in whom an ICD had been implanted based on an outdated risk assessment, have limited indications of ICD replacement. Still, a clinical decision to implant an ICD or replace it is straightforward in high-risk patients, but difficult in lower-risk patients [11]. It may be necessary to try and estimate the risk more precisely than current guidelines allow us to.

To date, the American guidelines are based on classical risk factors, which means that the SCD risk before ICD replacement, in most cases, is regarded at least as high as at the first implantation [8]. On the other hand, the European guidelines acknowledge that the risk of SCD changes over time, requiring that individual risk should be reassessed before each device replacement. In addition, the procedural risk (different for single battery replacement and lead extraction) should be taken into account in order to carefully weigh the benefits and risks of intervention [12,13].

Therefore, it seems we need a large multicenter registry to validate an algorithm that justifies the risk of ICD replacement in low SCD risk patients with sufficient specificity. Present recommendations only take into account the patient’s age [6,8]. Some young HCM patients having initially high calculated risk, with a stable course of the disease, may move over time into a lower-risk group—their indications of ICD may expire. For some suitable patients, a solution could be a completely extracardiac system—a subcutaneous ICD implanted only for a limited time, when the risk of SCD seems to be the highest. On the other hand, HCM is regarded as a progressive disease associated with the progression of myocardial fibrosis [14,15]. Of note, the extent of myocardial fibrosis is not included in the SCD risk calculator advocated by European guidelines.

Some limitations of our study must be considered. Despite its limitations and less than universal acceptance, in order to quantify the SCD risk, we used the ESC-advocated risk calculator. It is used in our center as a tool we refer to, although, of course, full individual patient risk assessment encompassing more factors than those included in the calculator is always performed. However, as the score gives a numerical risk range, as with any score or index, it is invaluable for quantification purposes. The size and character of the study make it appropriate only to describe the clinical problem, however, it does not allow for addressing it in a significant way. For that, larger, differently designed studies are needed.

## 5. Conclusions

As the precision of the SCD risk assessment in HCM patients has improved, there have been evolutionary changes in the guidelines, impacting the evaluation of patients’ SCD risk. Our study shows that the risk of patients referred for an ICD placement years ago tended to be lower than that of patients in whom currently a device placement is indicated. Moreso, the risk of SCD in HCM may change over time, sometimes becoming lower. Patients, therefore, require reassessment prior to exchanging the device. Optimal treatment strategy in such cases, which constitutes a growing proportion of HCM patients, needs to be evaluated in multicenter studies and registries of appropriate size. We need more data, based on more factors than the ESC SCD calculator, allowing us to quantify the risk of SCD in HCM patients. As things currently stand, we may be overestimating this risk in some patients, which could lead to unnecessary morbidity and cost related to an increased number of ICD implantations.

## Figures and Tables

**Figure 1 jcm-11-01633-f001:**
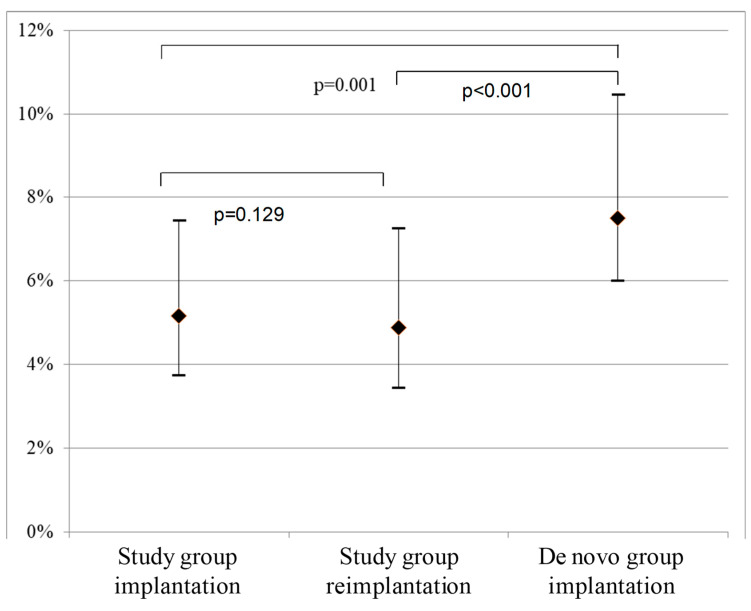
Calculated 5-year SCD risk (median and IQR) in the study group and de novo group.

**Table 1 jcm-11-01633-t001:** Major risk factors and calculated risk of SCD at 5 years in patients with appropriate ICD intervention in a group of patients implanted with ICD for primary prevention.

	Age	Risk of SCD at 5 Years (%)	Max. (mm) LVWT	FHSCD	Syncope	nsVT	ABPR
1	36	9.29	15	No	Yes	Yes	No
2	21	6.76	35	No	No	Yes	Yes
3	57	6.65	19	No	Yes	Yes	No
4	51	NA *	36	No	No	No	No
5	46	11.36	21	Yes	Yes	No	Yes
6	45	11.74	31	No	No	Yes	Yes
7	21	11.00	24	No	No	Yes	Yes
8	53	7.97	30	No	No	Yes	No
9	31	NA **	20	No	No	Yes ***	No
10	51	12.78	24	No	Yes	Yes	No
11	21	12.53	32	No	Yes	Yes	No
12	46	15.35	22	Yes	Yes	Yes	Yes
13	46	9.08	22	Yes	No	Yes	No
14	36	6.53	24	No	No	Yes ****	No

SCD—sudden cardiac death; LVWT—left ventricular wall thickness; FHSCD—family history of sudden cardiac death; nsVT—no-sustained ventricular tachycardia, ABPR—abnormal blood pressure reaction during the exercise test, * LVWT > 35 mm (risk cannot be calculated), ** after myectomy with mitral valve replacement (risk cannot be calculated), *** fast, long nsVT with presyncope, **** nsVT during the exercise test.

## Data Availability

Not applicable.

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
