# Peer review of "Sudden Cardiac Death Risk over Time in HCM Patients with Implantable Cardioverter-Defibrillator"

_jcm, 2022, doi:10.3390/jcm11061633_

Round 1

Reviewer 1 Report

A very good topic, but I don´t understand the selection of the patients- in the section Materials and methods- 91 patients with ICD implantation between 2005-2013- in the section results- 70 patients first ICD, 21 second or third..

It is not clear the number of patients in the groups...

Which score was used for the re- implantation group??

in the table 1- NA?? why??

I don´t understand tab. 2

Author Response

I would like to thank the Reviewer for in-depth paper analysis and helpful suggestions. Below I have enumerated and am commenting on the amendments made in response to Reviewer's comments.

A very good topic, but I don´t understand the selection of the patients- in the section Materials and methods- 91 patients with ICD implantation between 2005-2013- in the section results- 70 patients first ICD, 21 second or third..It is not clear the number of patients in the groups...

Thank you for this comment. We included an initial cohort of 91 patients, but over time some of them underwent procedures (eg. myectomy) or had events (VT, VF) that excluded them from repeat analysis with the ESC calculator. An additional sentence of explanation is added in lines 76-77. This is elucidated further in section 2.1 Risk calculation and comparisons.

Which score was used for the re- implantation group??

The same ESC 2014 risk calculator was used for the re-implantation cohort.

in the table 1- NA?? why??

The NA in Table 1 is explained in the legend under the double asterisk reference - again, the calculator cannot be used in certain patients as it has not been validated in some settings.

I don´t understand tab. 2

Thank you for this comment. Table 2 is indeed unclear and partially redundant as most of its contents is also presented in the text. We removed it altogether. The paragraph describing its contents was corrected accordingly.

Reviewer 2 Report

This is generally a well-written and comprehensive article that aimed to investigate how the calculated SCD risk in HCM patients with an ICD implanted in the past changed over time. The article is well-structured, very interesting and the results are presented in an appropriate manner, being clear and transparent. The statistical analysis is also well done. Even though this study was retrospective and included patients from one cardiology center, I consider that the findings are interesting and that the results obtained can make significant contributions to further large studies. I consider that the study is valuable and sound and can be published after some minor revisions. I suggest paying attention to the English language and correcting some missing letters. Also, I suggest addressing the future scope and topics that are important and that could not be covered in the manuscript.

Author Response

I would like to thank the Reviewer for in-depth paper analysis and helpful suggestions.

Below I have enumerated and am commenting on the amendments made in response to Reviewer’s comments

This is generally a well-written and comprehensive article that aimed to investigate how the calculated SCD risk in HCM patients with an ICD implanted in the past changed over time. The article is well-structured, very interesting and the results are presented in an appropriate manner, being clear and transparent. The statistical analysis is also well done. Even though this study was retrospective and included patients from one cardiology center, I consider that the findings are interesting and that the results obtained can make significant contributions to further large studies. I consider that the study is valuable and sound and can be published after some minor revisions. I suggest paying attention to the English language and correcting some missing letters.

Thank you for this comment. The corrections have been made.  

Also, I suggest addressing the future scope and topics that are important and that could not be covered in the manuscript.

Future scope and topics were added: We need more data, based on more factors than the ESC SCD calculator, allowing us to quantify the risk of SCD in HCM patients. As things currently stand, we may be overestimating this risk in some patients, which could lead to unnecessary morbidity and cost related to an increased number of ICD implantations.

Round 2

Reviewer 1 Report

accept